# Genome-Wide Differential Transcription of Long Noncoding RNAs in Psoriatic Skin

**DOI:** 10.3390/ijms242216344

**Published:** 2023-11-15

**Authors:** Valerie M. Stacey, Sulev Kõks

**Affiliations:** 1Perron Institute for Neurological and Translational Science, 8 Verdun Street, Nedlands, WA 6009, Australia; valerie.m.stacey.26@dartmouth.edu; 2Centre for Molecular Medicine and Innovative Therapeutics, Murdoch University, Perth, WA 6150, Australia

**Keywords:** psoriasis, long noncoding RNAs (lncRNAs), differential expression, transcription

## Abstract

Long noncoding RNAs (lncRNAs) may contribute to the formation of psoriatic lesions. The present study’s objective was to identify long lncRNA genes that are differentially expressed in patient samples of psoriasis through computational analysis techniques. By using previously published RNA sequencing data from psoriatic and healthy patients (n = 324), we analysed the differential expression of lncRNAs to determine transcripts of heightened expression. We computationally screened lncRNA transcripts as annotated by GENCODE across the human genome and compared transcription in psoriatic and healthy samples from two separate studies. We observed 54 differentially expressed genes as seen in two independent datasets collected from psoriasis and healthy patients. We also identified the differential expression of LINC01215 and LINC1206 associated with the cell cycle pathway and psoriasis pathogenesis. SH3PXD2A-AS1 was identified as a participant in the STAT3/SH3PXD2A-AS1/miR-125b/STAT3 positive feedback loop. Both the SH3PXD2A-AS1 and CERNA2 genes have already been recognised as part of the IFN-γ signalling pathway regulation. Additionally, EPHA1-AS1, CYP4Z2P and SNHG12 gene upregulation have all been previously linked to inflammatory skin diseases. Differential expression of various lncRNAs affects the pathogenesis of psoriasis. Further characterisation of lncRNAs and their functions are important for developing our understanding of psoriasis.

## 1. Introduction

Psoriasis is a chronic and inflammatory skin disease that presents in multiple forms and affects an estimated 60 million people worldwide, though estimates of prevalence range from between 0.09 and 11.4% of the population [1,2,3]. Psoriasis causes both individual lifestyle and wider-scale economic liabilities. The estimated annual healthcare burden of psoriasis in the United States is estimated to be around USD 35.2 billion as of 2016 [4]. It is also associated with instances of depression and anxiety amongst adults [5]. Psoriasis is associated with a strong genetic predisposition; however, the specific cause is unknown [3,6].

The pathophysiology of psoriasis is complex and involves the interaction of genetic and environmental factors. It is a T-cell-mediated autoimmune disease similar to other complex autoimmune disorders such as rheumatoid arthritis, Crohn’s disease and diabetes. Psoriasis resembles other autoimmune disorders by several criteria: distinct role of genetic and environmental factors, variable age of onset, the significant variability of the tissue reaction with different degrees of activity and frequency of relapses [7]. Primarily taken as skin disorder, it is now known that psoriasis is a chronic systemic disorder affecting other systems like the cardiovascular system and metabolism as well. Almost half of patients with psoriasis have psoriatic arthritis that is usually seronegative for rheumatoid factor and is presented in several characteristic forms [7].

Family studies have suggested that psoriasis has a complex mode of inheritance involving the interaction of multiple genes in combination with environmental factors. Concordance rate in monozygotic twins is a substantial 70%. A recent genome-wide association study (GWAS) identified 109 distinct psoriasis susceptibility loci [8]. In addition to genetic factors, epigenetic modifiers and mechanisms have a significant impact on the keratinocyte differentiation and therefore on the pathogenesis of psoriasis [9].

The unifying hypothesis of disease pathophysiology is the cytokine network model. In this model, either an exogenously derived stimulus, such as trauma, or an endogenous stimulus, such as HIV-1, neuropeptides, or medications, is portrayed as triggering a plexus of cellular events by inciting a cascade of cytokines. This model featured central pathogenetic roles for TNF-a derived from dendritic APCs and IFN-g produced by activated T helper 1/T cytotoxic 1 (Th1/Tc1)-type lymphocytes. These cytokines stimulate inflammation in the Type 1 pathway, increasing transcription of other T-helper 1 genes, and enhance the production of pro-inflammatory cytokines by keratinocytes, dendritic cells (DCs) and T-cells [7]. The cytokine network model also elucidates the epigenetic mechanisms behind the environmental factors triggering the disease [9].

Microarray and RNA sequencing profiling techniques have been used to investigate psoriatic cells [10] and have highlighted a large number of differential genes in lesional and non-lesional skin [11]. Transcriptome profiling is commonly used for generating hypotheses regarding molecular pathways associated with psoriasis as well as the development of potential therapeutic targets [12]. 

lncRNAs are a prevalent and functionally diverse class of non-coding RNAs—a group of RNAs that do not encode information about proteins [11,12,13,14,15]. lncRNAs are most commonly defined as RNAs that are over 200 nucleotides in length [15,16]. Evidence suggests that lncRNAs play a significant role in a number of biological processes [15,17,18], including transcription through epigenetic regulation [15,19], protein localisation [15,20,21], cell signalling [22], cell structure [23], the cell cycle [24], apoptosis [25] and the development of many human diseases, specifically autoimmune and inflammatory diseases [26,27,28,29]. New studies are also beginning to detail how lncRNAs play a part in the emergence of psoriasis [29,30,31]. Results have shown that lncRNAs are involved in the genesis and progression of psoriasis, specifically the inflammation levels [32]. These developments are important for understanding the genetic background of disease [33]. Through identifying clinical biomarkers, personalised medicine and gene therapy for psoriasis may become feasible and effective treatment methods [34]. However, little is known about the specific functions of certain novel lncRNAs due to lncRNAs being a relatively new area of research [31]. Numerous lncRNAs with unknown functions have been discovered in various human cell types [35]. A similar study previously conducted by Gupta et al. used a sample size of 18 psoriasis patients and 16 healthy controls. Results revealed 971 differentially expressed lncRNAs between pre-treatment psoriasis patients and healthy controls. Eleven lncRNAs were found to be co-expressed with protein-coding genes [36]. To our knowledge, there have been no comprehensive studies across the entire transcriptome using multiple RNA sequence datasets from different independent studies. The goal of this study is to build upon prior research and identify a more comprehensive list of differentially expressed lncRNAs in psoriasis lesions. This study has analysed publicly available previously sequenced RNA data from two large independent studies in order to compare lncRNA expression in psoriatic and healthy cells. One study had 177 and the other had 147 skin biopsy samples from healthy and psoriatic lesions. All samples were from psoriasis vulgaris patients, and the samples were homogenous.

## 2. Results

### 2.1. GSE54456 Analysis

A total of 16,582 lncRNA genes were shared between healthy and lesional cells in the GSE54456 dataset. There were a total of 4481 differentially expressed genes, defined as genes expressed at a FDR value of less than 0.05. Table 1 provides the 20 most significant differentially expressed genes. The genes with the lowest FDR values also had the largest absolute logFC values.

The most significantly upregulated gene is CERNA 2, a competing endogenous lncRNA 2 for microRNA let-7b. This is followed by LOC122526782, an uncharacterised lncRNA. The third most upregulated gene is ENSG00000285974 and is yet to be characterised. The only significantly downregulated gene was ENSG00000273132, which has not been characterised.

### 2.2. GSE121212 Analysis

In the GSE121212 sample, 13,742 lncRNA genes were shared between healthy and non-lesional cells. First, healthy cells were compared with psoriatic but non-lesional cells. There were a total of 73 differentially expressed genes with an FDR of less than 0.05. Table 2 lists the 20 most significantly differentially expressed genes. The most highly upregulated was LOC124901427, an uncharacterised lncRNA. Following this is the EPHA1-AS1 gene, which is an EPHA1 antisense RNA1. This is followed by the LOC122526782 gene, which is also an uncharacterised lncRNA. The most significantly downregulated gene was LOC107986954, an uncharacterised lncRNA. The second most highly downregulated gene was ENSG00000231204, which has not been characterised.

When comparing the psoriatic lesional and healthy skin biopsies in the GSE121212 sample, results showed 13,685 lncRNAs were expressed. Our analysis identified 2277 differentially expressed genes that had FDR values of less than 0.05. Table 3 shows the 20 most significantly differentially expressed genes ordered by ascending FDR values. The most highly expressed gene was ENSG00000253161, an uncharacterised lncRNA. The second most upregulated gene is LOC12491427, an uncharacterised lncRNA. The third most highly upregulated gene is ENSG00000287426, another uncharacterised lncRNA. The most highly downregulated gene was LINC02747, a long intergenic non-protein-coding RNA 2747. This is followed by the gene ENSG00000272666, a gene that has not been characterised. The third most significantly downregulated gene was ENSG00000287563, another uncharacterised gene.

### 2.3. General Analysis

The number of differentially expressed lncRNAs between lesional and healthy skin that were identical in both datasets was 1673. All differentially expressed genes seen across the GSE4456 data, and the two analyses of the GSE121212 transcription data can be seen in Table 4. Altogether, there are 57 differentially expressed genes that are shared across both studies and all three conditions of healthy, non-lesional and lesional skin. However, only 54 showed the same direction of regulation and were further studied. Of these, 38 lncRNA genes are upregulated, and 16 lncRNA genes are downregulated. Figure 1 shows the differential expression of these genes as a heatmap based on logFC values, illustrating how upregulated or downregulated each lncRNA gene is. It shows more significant differential expression when healthy and lesional samples are compared, and only moderate differential expression when healthy and non-lesional samples are compared. Of these genes, 28 of them are yet to be characterised, and their functions are completely unknown.

Figure 2 is a series of box plots highlighting gene expression of significant differentially expressed genes. The box plots compare the expression between lesional and psoriatic samples. The six genes that were chosen were all a part of the 20 most differentially expressed genes in both the GSE54456 dataset and the GSE121212 dataset and represent the most highly differentially expressed genes across both studies. Results show that CERNA2, LOC122526782, LOC124901427, LINC01215, CT69 and SH3PXD2A-AS1 all show moderate differential expression in non-lesional psoriatic samples and significantly higher differential expression in lesional psoriatic samples.

## 3. Discussion

In the present study, we investigated the lncRNA transcriptomes of the GSE54456 and GSE121212 RNA sequences for differentially expressed lncRNA genes in healthy, non-lesional psoriatic skin and lesional psoriatic skin. This resulted in 54 differentially expressed lncRNA genes observed across all conditions.

Of the differentially expressed genes observed through computational analysis, eight lncRNA genes have been associated with psoriasis or other inflammatory skin diseases in previous studies. EPHA1-AS1 has been noted as differentially expressed in the GSE13355 dataset as seen in a prior psoriasis study [37]. Despite this, little is known about its function in psoriasis, and its upregulation was not further analysed. 

The CYP4Z2P gene was also found to be differentially expressed in lesional cells, as seen in prior studies [30]. This Song et al. study suggested that this gene acts as a biomarker for psoriasis. The study used similar computational techniques to analyse a prior study conducted by Hangauer et al. [30,38]. qPCR and RNA sequencing were both conducted during this study, and CYP4Z2P, TRHDE-AS1 and HINT1 were listed as differentially expressed in skin lesions. However, our results did not highlight either TRHDE-AS1 or HINT1 as being differentially expressed. This difference may be due to the separate datasets being referenced, along with the Hangauer et al. study utilising qPCR in addition to RNA sequencing to determine the genes being differentially expressed.

The SNHG12 gene was also observed at elevated levels in chronically inflamed skin, seen in a study conducted by Liu et al. [39]. It was part of the CD3D+ T cell cluster found in the skin sample. Overall, this cluster was found to contain elevated levels of SNGH12, BATF and ZFAS1 lncRNAs. By contrast, our investigation only showed ZFAS1 upregulation in the GSE121212 dataset, and neither noted significantly differentially expression of BATF lncRNAs. These differences may be in part due to the Liu et al. study utilising samples that comprised five separate inflammatory skin conditions, each of which would yield their own respective transcriptome. The study also specifically collected immune cells, whilst our investigation did not investigate a specific cell type. The function of SNHG12 has not been identified yet.

LINC01215 upregulation, which is seen in our investigation, has been linked to general atopic dermatitis, another inflammatory skin disease [40,41]. This study by Zhu et al. investigated seven datasets and noted LINC01215 as one of 51 significant differentially expressed genes. No further detail was discussed regarding its function. It has been reported that LINC01215 promotes epithelial–mesenchymal transition through RUNX3 promoter methylation [42].

The significantly upregulated LINC01215 and LINC1206 lncRNA genes as noted in Table 4 were both observed to interact with CCNA2, CCNB1 and CCNE1 factors, which are part of the cell cycle pathway to regulate psoriasis progression [42,43]. This study also pinpointed LINC01269 as another significant contributor to this pathway. Although this gene was not found to be differentially expressed in the GSE121212 dataset, it was found in the analysis of the GSE4456 dataset. 

SH3PXD2A-AS1 upregulation has been associated specifically with psoriasis in previous studies [44]. Yang et al. analysed the GSE50790 and GSE13355 microarray profiles and highlighted SH3PXD2A-AS1 as an important component of psoriasis progression. This lines up with our observed results. The upregulation of SH3PXD2A-AS1 was found to promote HaCaT cell proliferation and supress HaCaT cell apoptosis, participating in the STAT3/SH3PXD2A-AS1/miR-125b/STAT3 positive feedback loop which affects the pathogenesis of psoriasis [44]. 

CERNA2 is also highly upregulated in psoriatic skin. A study by Lin et al. suggested that CERNA2, SH3PXD2A-AS1 and PRKCQ-AS1 are positively correlated to STAT1 and may participate in the psoriasis progression through the regulation of the IFN-γ and JAK/STAT signalling pathways as well as inflammatory levels in human keratinocytes [31]. The upregulation of CERNA2 and SH3PXD2A-AS1 can be corroborated by both the GSE54456 and GSE121212 datasets. However, PRKCQ-AS1 was only seen to be differentially expressed in the GSE121212 dataset.

In conclusion, our approach to investigating lncRNA expression has highlighted 1673 lncRNA genes that seem to play a part in the development of psoriasis lesion. The 57 lncRNAs that were found to be overlapping between three different skin conditions (healthy, lesional and non-lesional) describe more general psoriasis susceptibility lncRNAs. We also corroborated the involvement of eight lncRNA genes involved in psoriasis as determined by previous studies, including EPHA1-AS1, CYP4Z2P, SNHG12, LINC01215, LINC1206, SH3PXD2A-AS1 and CERNA2. The field of research involving lncRNAs is still undergoing significant development, and there is substantial information that is yet to be uncovered regarding their involvement in psoriasis—as seen in the 29 lncRNA genes that are yet to be characterised. Only LINC01215, LINC1206, LINC1269, SH3PXD2A-AS1, CERNA2 and PRKCQ-AS1 have known functions and pathways associated with psoriasis pathogenesis. Future investigations into lncRNA functions can be conducted to identify more lncRNAs involved in psoriasis and their specific involvement in pathways.

## 4. Materials and Methods

### 4.1. Data Collection

RNA sequence datasets derived from healthy and psoriatic patients were downloaded from the NCBI Gene Expression Omnibus database. Sequences with accession numbers GSE54456 (PMID:24441097) [45] and GSE121212 (PMID:30641038) [46] were used. GSE54456 collected data from 95 psoriatic and 84 normal skin samples (total 179), whilst GSE121212 collected 55 samples from lesional skin and 54 from non-lesional skin of psoriasis patients and 38 samples from the skin of healthy controls. GSE54456 samples were classified into healthy and lesional skin, whilst GSE121212 samples were classified into healthy, non-lesional, lesional and chronically lesioned skin.

The lncRNA annotations were downloaded from the GENCODE database; release 43 was used. The dataset contained sequences for 58,032 lncRNA transcripts annotated by GENCODE. This reference sequence corresponds to the hg38 (GRCh38.p13) human reference genome.

### 4.2. Computational Analysis

Differentially expressed genes were detected by comparing psoriasis and healthy groups using DEseq2 [47]. Genes were then annotated using BiomaRt [48]. Differentially expressed genes were identified using *p*-value < 0.05. The most highly upregulated genes were selected for further analysis. Zenbu was used to identify the transcripts of each gene that was differentially expressed [49]. ggpubr was then used to generate box plots to visualise the difference in expression between healthy and psoriatic samples.

## 5. Conclusions

In this study, we identified several lncRNAs to be differentially expressed in psoriasis samples, and various lncRNAs affect the pathogenesis of psoriasis. Further functional characterisation of lncRNAs is important to understand their molecular role in the development of psoriasis.

## Figures and Tables

**Figure 1 ijms-24-16344-f001:**
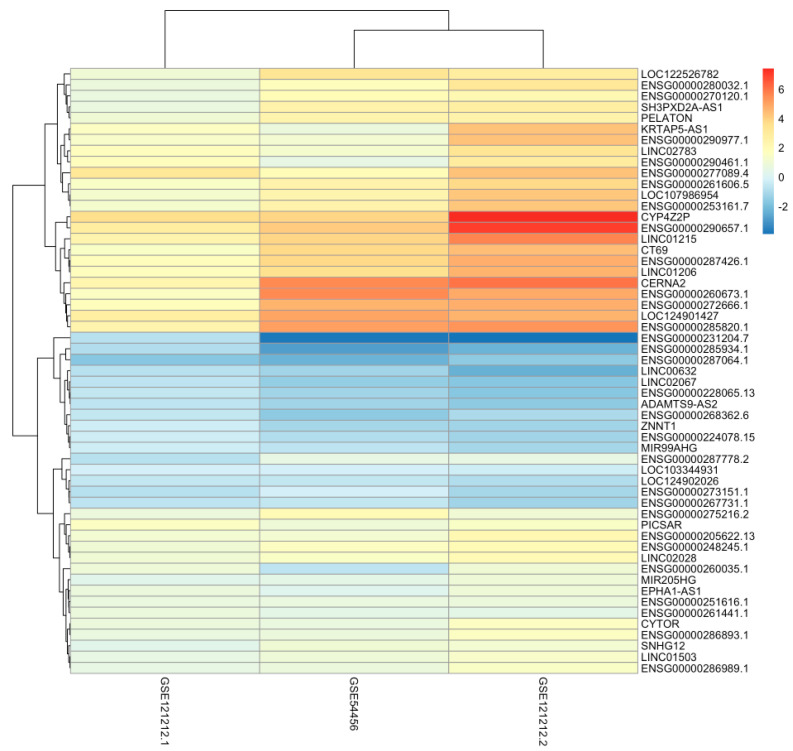
This heatmap shows the differential expression of the 54 lncRNA genes that were identified to be differentially expressed across all three conditions (healthy, non–lesional and lesional). It offers a visualisation of statistically significant lncRNA genes that play a part in the pathogenesis of psoriasis.

**Figure 2 ijms-24-16344-f002:**
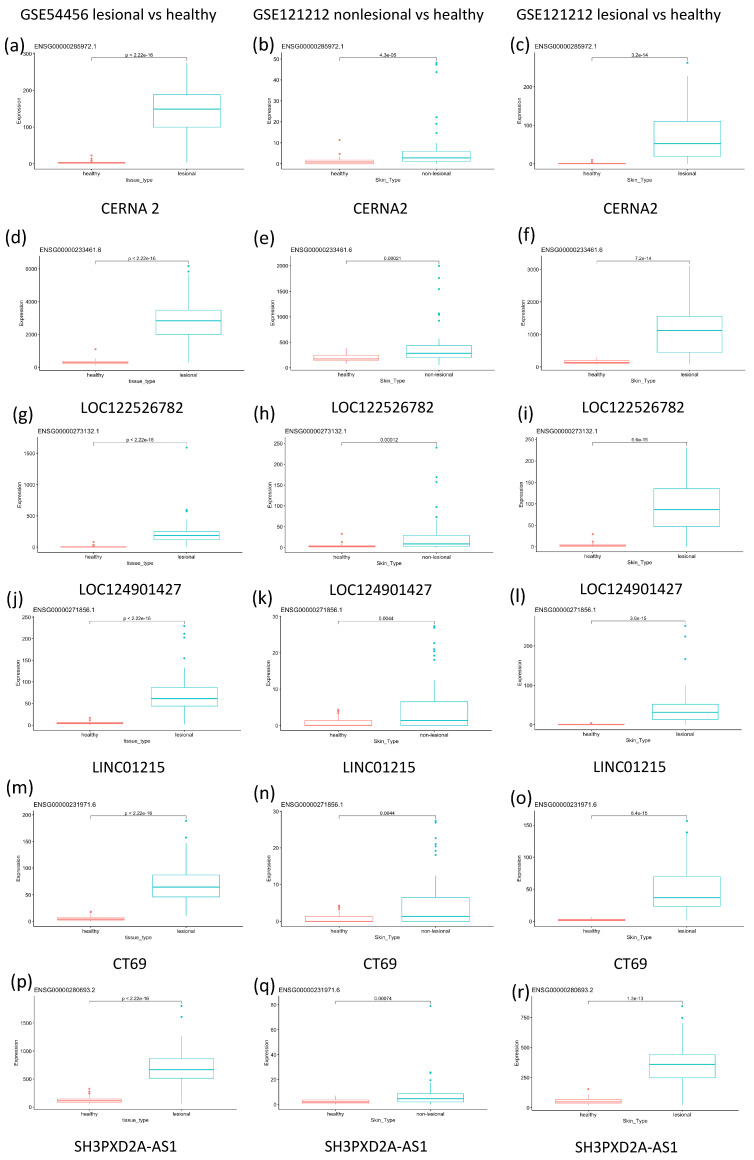
This figure shows box plots of the expression of six significant differentially expressed genes. (**a**) CERNA2 gene expression in healthy vs. lesional skin as observed in the GSE54456 dataset; (**b**) CERNA 2 gene expression in healthy vs. non–lesional skin as observed in the GSE121212 dataset; (**c**) CERNA 2 gene expression in healthy vs. lesional skin as observed in the GSE121212 dataset; (**d**) LOC122526782 gene expression in healthy vs. lesional skin as observed in the GSE54456 dataset; (**e**) LOC122526782 gene expression in healthy vs. non–lesional skin as observed in the GSE121212 dataset; (**f**) LOC122526782 gene expression in healthy vs. lesional skin as observed in the GSE121212 dataset; (**g**) LOC124901427 gene expression in healthy vs. lesional skin as observed in the GSE54456 dataset; (**h**) LOC124901427 gene expression in healthy vs. non–lesional skin as observed in the GSE121212 dataset; (**i**) LOC124901427 gene expression in healthy vs. lesional skin as observed in the GSE121212 dataset; (**j**) LINC01215 gene expression in healthy vs. lesional skin as observed in the GSE54456 dataset; (**k**) LINC01215 gene expression in healthy vs. non–lesional skin as observed in the GSE121212 dataset; (**l**) LINC01215 gene expression in healthy vs. lesional skin as observed in the GSE121212 dataset; (**m**) CT69 gene expression in healthy vs. lesional skin as observed in the GSE54456 dataset; (**n**) CT69 gene expression in healthy vs. non–lesional skin as observed in the GSE121212 dataset; (**o**) CT69 gene expression in healthy vs. lesional skin as observed in the GSE121212 dataset; (**p**) SH3PXD2A-AS1 gene expression in healthy vs. lesional skin as observed in the GSE54456 dataset; (**q**) SH3PXD2A-AS1 gene expression in healthy vs. non–lesional skin as observed in the GSE121212 dataset; (**r**) SH3PXD2A-AS1 gene expression in healthy vs. lesional skin as observed in the GSE121212 dataset.

**Table 1 ijms-24-16344-t001:** Analysis of differentially expressed genes in psoriasis seen in the GSE54456 sample, sorted by FDR value. Differentially expressed genes in lesional psoriatic cells compared to healthy skin cells. logFC describes how many times gene expression differs between the two groups and is presented on the log2 scale. Positive values suggest upregulation, and negative values suggest downregulation. FDR is the genome-wide corrected *p*-value.

Gene	logFC	*p*-Value	FDR	Symbol	Gene Name
ENSG00000285972.1	5.54	0	0	CERNA2	competing endogenous lncRNA 2 for microRNA let-7b
ENSG00000233461.6	3.26	1.72 × 10^−271^	9.03 × 10^−268^	LOC122526782	uncharacterised LOC122526782
ENSG00000260673.1	5.60	4.93 × 10^−245^	1.73 × 10^−241^	lnc-RPP40-3	NA
ENSG00000214999.3	2.56	1.32 × 10^−237^	3.47 × 10^−234^	ALOX12-AS1	ALOX12 antisense 1
ENSG00000280693.2	2.54	6.90 × 10^−197^	1.45 × 10^−193^	SH3PXD2A-AS1	SH3PXD2A antisense RNA 1
ENSG00000271856.1	3.87	3.29 × 10^−195^	5.76 × 10^−192^	LINC01215	long intergenic non-protein-coding RNA 1215
ENSG00000272666.1	4.61	1.20 × 10^−189^	1.80 × 10^−186^	KLHDC7B-DT	KLHDC7B Divergent Transcript
ENSG00000231971.6	3.73	4.45 × 10^−187^	5.84 × 10^−184^	CT69	cancer/testis associated transcript 69
ENSG00000273272.2	4.41	4.97 × 10^−164^	5.81 × 10^−161^	NA	NA
ENSG00000286962.1	4.34	1.61 × 10^−162^	1.70 × 10^−159^	SLC6A11-AS1	antisense to SLC6A11
ENSG00000253417.5	3.69	3.41 × 10^−142^	3.26 × 10^−139^	LINC02159	long intergenic non-protein-coding RNA 2159
ENSG00000273132.1	5.01	1.59 × 10^−135^	1.39 × 10^−132^	LOC124901427	uncharacterised LOC124901427
ENSG00000285820.1	5.05	6.36 × 10^−127^	5.14 × 10^−124^	ASTN2-AS1	antisense to ASTN2
ENSG00000251320.1	−2.78	2.97 × 10^−126^	2.23 × 10^−123^	NA	NA
ENSG00000275880.1	1.68	1.19 × 10^−123^	8.35 × 10^−121^	NAXD-AS1	NAXD antisense RNA 1
ENSG00000272711.1	1.97	1.89 × 10^−119^	1.24 × 10^−116^	HK2-DT	HK2 Divergent Transcript
ENSG00000254027.1	3.32	4.02 × 10^−117^	2.49 × 10^−114^	LNMICC	NA
ENSG00000258689.2	5.27	1.11 × 10^−115^	6.45 × 10^−113^	LINC01269	long intergenic non-protein-coding RNA 1269
ENSG00000245648.2	2.41	7.96 × 10^−113^	4.40 × 10^−110^	KLRK1-AS1	KLRK1 antisense RNA 1
ENSG00000288575.1	3.62	1.16 × 10^−110^	6.08 × 10^−108^	NA	NA

**Table 2 ijms-24-16344-t002:** Analysis of differentially expressed genes in non-lesional psoriasis samples seen in the GSE121212 sample, sorted by FDR value. Differentially expressed genes in non-lesional psoriatic cells compared to healthy skin cells. logFC describes how many times gene expression differs between the two groups and is presented on the log2 scale. Positive values suggest upregulation, and negative values suggest downregulation. FDR is the genome-wide corrected *p*-value.

Gene	logFC	*p*-Value	FDR	Symbol	Gene Name
ENSG00000273132.1	2.67	2.56 × 10^−10^	1.24 × 10^−6^	LOC124901427	uncharacterised LOC124901427
ENSG00000229153.7	0.81	1.09 × 10^−8^	2.63 × 10^−5^	EPHA1-AS1	EPHA1 antisense RNA 1
ENSG00000233461.6	1.03	4.20 × 10^−8^	5.32 × 10^−5^	LOC122526782	uncharacterised LOC122526782
ENSG00000285972.1	2.33	4.40 × 10^−8^	5.32 × 10^−5^	CERNA2	competing endogenous lncRNA 2 for microRNA let-7b
ENSG00000277089.4	3.13	1.89 × 10^−7^	1.83 × 10^−4^	CCL3-AS1	CCL3 antisense RNA 1
ENSG00000231971.6	1.52	6.17 × 10^−7^	4.97 × 10^−4^	CT69	cancer/testis associated transcript 69
ENSG00000204362.6	1.68	9.28 × 10^−6^	3.53 × 10^−3^	LINC02783	long intergenic non-protein-coding RNA 2783
ENSG00000234678.2	1.41	9.89 × 10^−6^	3.53 × 10^−3^	ELF3-AS1	ELF3 antisense RNA 1
ENSG00000234862.1	2.30	1.02 × 10^−5^	3.53 × 10^−3^	NA	NA
ENSG00000240401.9	1.12	7.07 × 10^−6^	3.53 × 10^−3^	NA	NA
ENSG00000260673.1	1.65	6.99 × 10^−6^	3.53 × 10^−3^	NA	NA
ENSG00000270120.1	0.71	9.73 × 10^−6^	3.53 × 10^−3^	NA	NA
ENSG00000271856.1	2.34	6.35 × 10^−6^	3.53 × 10^−3^	LINC01215	long intergenic non-protein-coding RNA 1215
ENSG00000272666.1	1.78	8.06 × 10^−6^	3.53 × 10^−3^	KLHDC7B-DT	KLHDC7B divergent transcript
ENSG00000287426.1	1.81	1.13 × 10^−5^	3.65 × 10^−3^	NA	NA
ENSG00000222041.13	0.82	1.29 × 10^−5^	3.91 × 10^−3^	CYTOR	cytoskeleton regulator RNA
ENSG00000203930.13	−0.76	1.43 × 10^−5^	4.01 × 10^−3^	LINC00632	long intergenic non-protein-coding RNA 632
ENSG00000287927.1	1.28	2.22 × 10^−5^	5.96 × 10^−3^	LOC107986954	uncharacterised LOC107986954
ENSG00000231204.7	−0.75	2.64 × 10^−5^	6.72 × 10^−3^	NA	NA
ENSG00000253161.7	1.24	2.87 × 10^−5^	6.95 × 10^−3^	LINC01605	long intergenic non-protein-coding RNA 1605

**Table 3 ijms-24-16344-t003:** Analysis of differentially expressed genes in lesional psoriasis samples seen in the GSE121212 sample, sorted by FDR value. Differentially expressed genes in lesional psoriatic cells compared to healthy skin cells. logFC describes how many times gene expression differs between the two groups and is presented on the log2 scale. Positive values suggest upregulation, and negative values suggest downregulation. FDR is the genome-wide corrected *p*-value.

Gene	logFC	*p*-Value	FDR	Symbol	Gene Name
ENSG00000253161.7	4.24	7.46 × 10^−82^	5.54 × 10^−78^	LINC01605	long intergenic non-protein-coding RNA 1605
ENSG00000273132.1	4.66	2.92 × 10^−70^	1.09 × 10^−66^	LOC124901427	uncharacterised LOC124901427
ENSG00000287426.1	4.74	2.26 × 10^−65^	5.59 × 10^−62^	LOC124902793	NA
ENSG00000255774.2	−2.60	3.26 × 10^−65^	6.06 × 10^−62^	LINC02747	long intergenic non-protein-coding RNA 2747
ENSG00000272666.1	4.73	1.03 × 10^−63^	1.53 × 10^−60^	KLHDC7B-DT	KLHDC7B divergent transcript
ENSG00000231204.7	−3.89	1.15 × 10^−61^	1.27 × 10^−58^	NA	NA
ENSG00000260673.1	4.83	1.20 × 10^−61^	1.27 × 10^−58^	NA	NA
ENSG00000280693.2	2.70	3.47 × 10^−59^	3.22 × 10^−56^	SH3PXD2A-AS1	SH3PXD2A antisense RNA 1
ENSG00000285972.1	5.97	1.74 × 10^−58^	1.43 × 10^−55^	CERNA2	competing endogenous lncRNA 2 for microRNA let-7b
ENSG00000231971.6	4.44	3.93 × 10^−57^	2.92 × 10^−54^	CT69	cancer/testis associated transcript 69
ENSG00000287563.1	−3.75	1.35 × 10^−56^	9.14 × 10^−54^	NA	NA
ENSG00000249859.14	1.66	1.12 × 10^−55^	6.91 × 10^−53^	PVT1	Pvt1 oncogene
ENSG00000280032.1	3.23	4.09 × 10^−54^	2.34 × 10^−51^	NA	NA
ENSG00000233461.6	2.85	8.99 × 10^−54^	4.77 × 10^−51^	LOC122526782	uncharacterised LOC122526782
ENSG00000233452.9	2.50	2.63 × 10^−51^	1.30 × 10^−48^	STXBP5-AS1	STXBP5 antisense RNA 1
ENSG00000271856.1	5.64	6.34 × 10^−51^	2.94 × 10^−48^	LINC01215	long intergenic non-protein-coding RNA 1215
ENSG00000253746.1	4.19	2.23 × 10^−49^	9.75 × 10^−47^	NA	NA
ENSG00000290657.1	7.08	8.04 × 10^−49^	3.32 × 10^−46^	NA	NA
ENSG00000223863.1	−1.79	2.64 × 10^−48^	1.03 × 10^−45^	LINC01805	long intergenic non-protein-coding RNA 1805
ENSG00000234678.2	3.45	5.81 × 10^−44^	2.1 × 10^−41^	ELF3-AS1	ELF3 antisense RNA 1

**Table 4 ijms-24-16344-t004:** Differentially expressed genes present in all three conditions: lesional, non-lesional and healthy skin. Genes listed are differentially expressed under three conditions—heathy, non-lesional and lesional skin. Gene name describes the function of the gene. Regulation states whether or not the gene has been upregulated or downregulated based on the FDR value.

Gene	Symbol	Gene Name	Regulation
ENSG00000273132.1	LOC124901427	uncharacterised LOC124901427	Upregulated
ENSG00000229153.7	EPHA1-AS1	EPHA1 antisense RNA 1	Upregulated
ENSG00000233461.6	LOC122526782	uncharacterised LOC122526782	Upregulated
ENSG00000285972.1	CERNA2	competing endogenous lncRNA 2 for microRNA let-7b	Upregulated
ENSG00000277089.4	CCL3-AS1	CCL3 antisense RNA 1	Upregulated
ENSG00000231971.6	CT69	cancer/testis associated transcript 69	Upregulated
ENSG00000204362.6	LINC02783	long intergenic non-protein-coding RNA 2783	Upregulated
ENSG00000260673.1	NA	NA	Upregulated
ENSG00000270120.1	NA	NA	Upregulated
ENSG00000271856.1	LINC01215	long intergenic non-protein-coding RNA 1215	Upregulated
ENSG00000272666.1	KLHDC7B-DT	KLHDC7B divergent transcript	Upregulated
ENSG00000287426.1	LOC124902793	uncharacterised LOC124902793	Upregulated
ENSG00000222041.13	CYTOR	cytoskeleton regulator RNA	Upregulated
ENSG00000203930.13	LINC00632	long intergenic non-protein-coding RNA 632	Downregulated
ENSG00000287927.1	LOC107986954	uncharacterised LOC107986954	Upregulated
ENSG00000231204.7	NA	NA	Downregulated
ENSG00000253161.7	LINC01605	long intergenic non-protein-coding RNA 1605	Upregulated
ENSG00000251616.1	RP11-485M7.3	NA	Upregulated
ENSG00000275874.1	PICSAR	P38 inhibited cutaneous squamous cell carcinoma associated lincRNA	Upregulated
ENSG00000228065.13	LINC01515	long intergenic non-protein-coding RNA 1515	Downregulated
ENSG00000230937.13	MIR205HG	MIR205 host gene	Upregulated
ENSG00000233901.8	LINC01503	long intergenic non-protein-coding RNA 1503	Upregulated
ENSG00000240567.1	LINC02067	long intergenic non-protein-coding RNA 2067	Downregulated
ENSG00000275216.2	LINC03061	long intergenic non-protein-coding RNA 3061	Upregulated
ENSG00000280693.2	SH3PXD2A-AS1	SH3PXD2A antisense RNA 1	Upregulated
ENSG00000248245.1	NA	NA	Upregulated
ENSG00000286893.1	NA	NA	Upregulated
ENSG00000286989.1	NA	NA	Upregulated
ENSG00000285820.1	ASTN2-AS1	antisense to ASTN2	Upregulated
ENSG00000261441.1	POLG-DT	POLG Divergent Transcript	Upregulated
ENSG00000290461.1	NA	NA	Upregulated
ENSG00000224397.9	PELATON	plaque enriched lncRNA in atherosclerotic and inflammatory bowel macrophage regulation	Upregulated
ENSG00000273151.1	NA	NA	Downregulated
ENSG00000290456.1	CYP4Z2P	cytochrome P450 family 4 subfamily Z member 2, pseudogene	Upregulated
ENSG00000267731.1	NA	NA	Downregulated
ENSG00000224078.15	SNHG14	small nucleolar RNA host gene 14	Downregulated
ENSG00000280032.1	NA	NA	Upregulated
ENSG00000215386.15	MIR99AHG	mir-99a-let-7c cluster host gene	Downregulated
ENSG00000261087.1	ZNNT1	ZNF706 neighbouring transcript 1	Downregulated
ENSG00000290657.1	NA	NA	Upregulated
ENSG00000268362.6	NA	NA	Downregulated
ENSG00000285934.1	NA	NA	Downregulated
ENSG00000230102.8	LINC02028	long intergenic non-protein-coding RNA 2028	Upregulated
ENSG00000260917.1	LOC103344931	uncharacterised LOC103344931	Downregulated
ENSG00000233930.4	KRTAP5-AS1	KRTAP5-1/KRTAP5-2 antisense RNA 1	Upregulated
ENSG00000287778.2	NA	NA	Downregulated
ENSG00000197989.15	SNHG12	small nucleolar RNA host gene 12	Upregulated
ENSG00000241684.8	ADAMTS9-AS2	ADAMTS9 antisense RNA 2	Downregulated
ENSG00000261606.5	NA	NA	Upregulated
ENSG00000287064.1	NA	NA	Downregulated
ENSG00000286535.1	LOC124902026	uncharacterised LOC124902026	Downregulated
ENSG00000205622.13	NA	NA	Upregulated
ENSG00000242512.9	LINC01206	long intergenic non-protein-coding RNA 1206	Upregulated
ENSG00000290977.1	NA	NA	Upregulated

## Data Availability

Data used in this study are available at Gene Expression Omnibus GSE54456 and GSE121212.

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
