# Peer review of "Genome-Wide Differential Transcription of Long Noncoding RNAs in Psoriatic Skin"

_ijms, 2023, doi:10.3390/ijms242216344_

Round 1
Reviewer 1 Report
Comments and Suggestions for Authors
In this work by V. M. Stacey and S. Kõks the authors report the results of bioinformatics analysis of RNA sequence datasets (GSE54456 and GSE121212) derived from healthy and psoriatic patients that downloaded from the NCBI Gene Expression Omnibus database. This work is one of an extensive series of works devoted investigation of potential links between transcriptome changes and psoriasis and generating hypotheses on molecular mechanisms associated with psoriasis.
Psoriasis is a chronic immune-mediated inflammatory dermatosis. The authors identify lncRNAs genes in healthy, non-lesional psoriatic skin and lesional psoriatic skin that differentially expressed in patient samples of psoriasis. As a result, in total 54 differentially expressed lncRNA genes observed across all conditions. Six genes (CERNA2, LOC122526782, LOC124901427, LINC01215, CT69 and SH3PXD2A-AS1) showed significantly higher differential expression in lesional psoriatic samples and moderate differential expression in non-lesional psoriatic samples. Involvement of SH3PXD2A-AS1, LINC01215 and CERNA2 in the development of psoriasis has been previously shown. Lnc SH3PXD2A-AS1 participates in TF STAT3 related mechanism STAT3/SH3PXD2A-AS1/miR-125b/STAT3 positive feedback loop that affects psoriasis pathogenesis via regulating human keratinocyte proliferation. LINC01215 is associated with cell cycle pathway and psoriasis pathogenesis. Cancer/testis antigen genes (CT69 gene in particular) encode a subgroup of tumor antigens expressed predominantly in testis and various tumors and is unlikely to related to the development of psoriasis. LOC122526782 and LOC124901427 are uncharacterized lncRNA.
It shown previously that lncRNAs might be potential biomarkers for psoriasis. In my opinion, the authors’ contribution in investigation of molecular mechanisms associated with psoriasis is relatively small.
Minor revision:
1. It is necessary to mention that LINC01215 promotes epithelial-mesenchymal transition through RUNX3 promoter methylation (Liu et al., 2021).
2. Line 152 – “Figure 2. This This figure…” - twice “this”.
Author Response
Dear reviewer
Thank you very much for your time and constructive criticism. We have evaluated yoru comments and modifed our mansucript accordingly MOre specifically:
It shown previously that lncRNAs might be potential biomarkers for psoriasis. In my opinion, the authors’ contribution in investigation of molecular mechanisms associated with psoriasis is relatively small.
Response: We added some information about the pathogenesis of psoriasis to the introduction and discussion. While some studies on the lncRNA and psoriasis already exist, our study used the largest publicly available datasets and provided confirmation to existing findings and also some new lncRNAs.
It is necessary to mention that LINC01215 promotes epithelial-mesenchymal transition through RUNX3 promoter methylation (Liu et al., 2021).
Response: We added the citation and sentence about this.
Line 152 – “Figure 2. This This figure…” - twice “this”.
Response: We removed the double word.
Thank you again for your constructive review. Hopefully this version is acceptable.
Reviewer 2 Report
Comments and Suggestions for Authors
The authors present a well-written article on the genome-wide differential transcription of long noncoding RNAs in psoriatic skin. The manuscript is well organized and the results provide valuable insights into this interesting topic.
Just two comments:
1) The authors should specify which type of psoriasis is being considered. Is the sample homogeneous?
2) Line 35-36: The authors should briefly discuss the entire pathogenetic scenario of psoriasis. Please consider all the factors involved in the pathogenesis of psoriasis with a special emphasis on the epigenetic landscape. This work is recent and could be useful: "doi: 10.3390/ijms23094874."
Author Response
Dear reviewer
Thank you very much for your time and effort. We very much appreciate your constructive comments.
The authors should specify which type of psoriasis is being considered. Is the sample homogeneous?
Response: We added information about sample homogeneity, All sampes were psoriasis vulgaris samples and were homogenous.
Line 35-36: The authors should briefly discuss the entire pathogenetic scenario of psoriasis. Please consider all the factors involved in the pathogenesis of psoriasis with a special emphasis on the epigenetic landscape. This work is recent and could be useful: "doi: 10.3390/ijms23094874."
Response: Yes, we considered the epigenetic changes and added appropriate refernces to the manuscript.
Hopefully we were able to improve our manuscript and address all your comments.